# Acute and Subchronic Oral Toxicity of Anthraquinone in Sprague Dawley Rats

**DOI:** 10.3390/ijerph191610413

**Published:** 2022-08-21

**Authors:** Jingjing Qu, Lanjie Pei, Xiangyan Wang, Shaohua Fu, Ling Yong, Xiao Xiao, Qianqian Xie, Bolin Fan, Yan Song

**Affiliations:** 1 Key Laboratory of Food Safety Risk Addessment, National Health and Family Planning Commission of the People’s Republic of China (China National Center for Food Safety Risk Assessment), Beijing 100022, China; 2 Hubei Provincial Key Laboratory for Applied Toxicology, Hubei Provincial Center for Disease Control and Prevention, Wuhan 430079, China

**Keywords:** anthraquinone, acute toxicity, subchronic toxicity, Sprague Dawley rats

## Abstract

Objective: This study was conducted to evaluate the acute and subchronic toxicity of anthraquinone. An acute toxicity test was performed in female Sprague Dawley (SD) rats, and the oral median lethal dose (LD_50_) of anthraquinone was estimated to be >5000 mg/kg body weight (BW). In the subchronic study, groups of 10 male and 10 female rats were dosed with anthraquinone by gavage at 0, 1.36, 5.44, 21.76, and 174.08 mg/kg BW, 7 days/week for 90 days followed by a recovery period of 28 days. No appreciable toxic-related changes were observed in the 1.36 mg/kg BW group. When the animals received 5.44 mg/kg BW or more of anthraquinone, hyaline droplet accumulation in the renal tubules was observed in both the male and female rats, and anemia was observed in the females. When the anthraquinone dose reached 174.08 mg/kg BW, mild hepatocellular hypertrophy around the central vein of the hepatic lobule and hypothyroidism were observed in the female rats. During the recovery period, changes in clinical symptoms and parameters were considerably alleviated. Based on the results of this study, the no observed adverse effect level (NOAEL) for anthraquinone in rats was set at 1.36 mg/kg BW, and the lowest observed adverse effect level (LOAEL) was 5.44 mg/kg BW.

## 1. Introduction

Anthraquinone (AQ), also known as 9,10-anthraquinone, is a derivative of anthracene. AQ can be detected in air, water, soil, plants, fish, seafood, and animal tissues because it can be generated from anthracene through various pathways, such as atmospheric oxidation, combustion processes, photolysis, and biodegradation [1,2,3]. In nature, AQs in plants (e.g., rhubarb) usually have certain pharmacological effects, including antibacterial, antitumor and pesticidal effects [4,5]. In industry, AQ is a raw material for the manufacture of many synthetic dyes and naturally occurring pigments and is also widely used as a raw material for the production of paper catalysts and hydrogen peroxide [6]. It has also been used as an active ingredient in pesticides and as a bird repellent but is now banned due to its stable nature and nondegradable properties [7,8].

In recent years, AQ residues exceeding 0.02 mg/kg have become the main cause of inspection failure in China’s tea exports to Europe [9]. According to some data, adult tea drinkers are exposed to 0.0026 μg/kg BW, people with high tea consumption are exposed to 0.0083 μg/kg BW, and the exposure ratio of male and female tea drinkers is 1.36:1. However, no relevant standards for the scope and limit of AQ in Chinese tea have been established in China. Therefore, it is necessary to identify the hazards of AQ for risk assessment in Chinese tea.

There are limited studies on the toxicity of AQ. The acute toxicity of AQ is low, with LD_50_ for the acute oral toxicity test of greater than 5000 mg/kg BW in rats and mice, an LD_50_ for acute intraperitoneal toxicity of greater than 3500 mg/kg BW, an LC_50_ for acute inhalation toxicity (4 h) of greater than 1.327 mg/L and an LD_50_ for the acute percutaneous toxicity test of greater than 500 mg/kg BW. The LD_50_ for the acute oral toxicity in rabbits is >3000 mg/kg BW, and the LD_50_ for the acute percutaneous toxicity test in rabbits is >3000 mg/kg BW [10,11]. The LD_50_ for the acute oral toxicity in bobwhite quail is >3000 mg/kg, and the LD_50_ for the acute oral toxicity in bluegill sunfish is >190 Fg/L [8]. The National Toxicology Program (NTP) has carried out 14-week feeding studies with AQ in F344/N rats and B6C3F1 mice. In these 14-week NTP studies, mild, responsive anemia was observed at doses of 1870 ppm (rat) and 3750 ppm (mouse) and starting at day 26 of the study. Additionally, altered renal function and thyroid gland follicular cell hypertrophy were apparent in the rats. Additionally, centrilobular hypertrophy of liver cells was observed in both the mice and rats at doses of 1875 ppm and above. All the exposed mice showed cytoplasmic alteration of the urinary bladder. Inflammation and transitional epithelial hyperplasia of the urinary bladder were observed in the female rats at 30,000 ppm. Hematopoietic cell proliferation and pigmentation in the spleens of the mice and rats corresponded to the hemolytic and regenerative nature of anemia [12]. Another article described a study wherein female F344 rats were exposed to AQ by dietary feed at concentrations of 0, 50, 150, 469, 938, 1875, or 3750 ppm for 13 weeks. The results showed that the mean body weight and food consumption for the rats in the 938 ppm group were 5–10% lower than the values for the rats in the control group. Increased liver, kidney, and spleen weights were observed in the rats exposed to AQ at 150 ppm and above. Increased bladder weights were observed from the 469 ppm group. Pathological findings were shown from 50 ppm AQ, spleen (mild hematopoietic cell proliferation and hyperpigmentation) and kidney (minimal clear droplets), and liver (mild lobular central hypertrophy) from the 938 ppm group. The conclusion of this article shows that there are no visible harmful effects at 469 ppm AQ (31.3 mg/kg BW); thus, this level was selected based on the absence of liver histopathology [13].

A comparison of the different AQ subchronic toxicity tests revealed a few problems with each of these studies. In the 14-week toxicity test by the NTP, no adverse effect level (NOAEL) was observed; in the Dodd et al. test, only female rats were used in the study, and the NOAEL was based on pathological findings in the liver, but pathological changes in the kidney and spleen were also observed in all dose groups. Therefore, the NOAEL of subchronic toxicity of AQ is uncertain, which is a key issue to be solved in this study. To clarify the 90-day NOAEL for AQ, we performed a subchronic toxicity test for AQ, to which we also added an acute toxicity test as the purity of AQ was different from the previously reported values.

## 2. Materials and Methods

The tests described in the following sections complied with the Organization for Economic Co-operation and Development (OECD) [14,15] Guidelines for Testing of Chemicals, the Chinese National Standard for Food Safety [16] and the laboratory’s good laboratory practice (GLP) code.

### 2.1. Test Materials

AQ (CAS No. 84-65-1) was obtained from Sigma-Aldrich (Product Number: A9004, Batch number: BCBT2720). A certificate of analysis from Sigma-Aldrich indicated a purity of 97%.

### 2.2. Animals and Living Conditions

This study was conducted at the Hubei Provincial Academy of Preventive Medicine in a GLP-certified laboratory. Male and female specific-pathogen-free (SPF) Sprague Dawley (SD) rats were obtained from the Hubei Provincial Center of Experimental Animals (Wuhan, China) (license number: SCXK(E)2015-0018). SD rats are outbred rats that are commonly used in pharmacological, toxicological, and pharmacodynamic studies. The animals were housed with 2 rats per individually ventilated cage (IVC) (40 air exchanges/h) in an environmentally controlled breeding room (temperature of 20–26 °C humidity of 40–70%) illuminated by artificial light with a 12 h light/dark cycle. Diet and drinking water were supplied in unlimited amounts.

### 2.3. Acute Oral Toxicity Study

Acute toxicity studies according to standardized protocols Developed by the OECD guideline 423 (Adopted: 2001) [14].

An acute toxicity test was performed in 6 female SD rats (three animals per step, two steps) with body weights ranging from 180 to 220 g. According to the information provided by Sigma-Aldrich, the test article showed an acute oral toxicity test LD_50_ > 2000 mg/kg BW in female rats. The starting dose was selected as 2000 mg/kg BW. After being fasted overnight, 3 rats were gavaged 2 times within 24 h with 10 mL/kg BW, with an interval of 4 h. The animals were observed individually for 30 min and once daily until the end of the observation period of 14 days. Another 3 animals were gavaged the same dose if no animals or just 1 animal died during the observation period. Alternatively, another 3 animals were tested at a lower dose. Mortality and toxic reactions were recorded every day, and the body weight was measured once weekly. At the end of the study, all the rats that survived were anesthetized with pentobarbital sodium and then sacrificed for gross anatomy. Then, we examined the head and neck, limbs, thoracic and abdominal organs of rats for signs of abnormalities and took samples of suspected and obvious lesions for histological examination.

### 2.4. Subchronic Oral Toxicity Study

#### 2.4.1. Animal Assignment, Dose and Formulation Analysis

Acute toxicity studies according to standardized protocols Developed by the OECD guideline 408 (adopted: 2018) [15] and the Chinese National Standard for Food Safety [16].

The subchronic toxicity test was performed in 120 SD rats of both sexes with an average body weight ranging from 60 to 80 g. After a 5-day acclimation, the animals were randomly assigned to 5 groups. Males and females with 10 rats/sex/group were dosed at 0.0, 1.36, 5.44, 21.76, and 174.08 mg/kg BW AQ AQ were solubilized in 1% sodium carboxymethylcellulose solution in order to obtain solutions of 0.136, 0.544, 2.176, and 17.408 mg/mL. Carboxymethyl cellulose sodium (1%) was used as the control. The gastric volume of the rats was 10 mL/kg BW. Additional satellite groups, with 5 rats/sex/group in the control and 174.08 mg/kg BW groups, were observed after a treatment-free recovery period of 28 days. This dosage design was based on the data provided by Sigma-Aldrich and previous AQ subchronic toxicity tests.

The stability of the AQ preparations was confirmed before the start of the study, and the homogeneity and concentration of the AQ preparations were tested in the study. The preparation samples were taken for analysis on the day of preparation. Two aliquots (one for testing and one for backup, 1 mL each) were taken from the top, middle, and bottom of the suspension and placed in labeled EP tubes. One hundred microliters of each sample was pipetted precisely into a 5 mL EP tube, to which 2400 μL of chloroform (analytical purity, Sinopharm Chemical Reagent Co., Ltd., Shanghai, China) was added for extraction by vortexing for 5 min on a vortex shaker; the lower layer of chloroform solution was diluted and mixed with methanol:water (90:10, *v*:*v*) and then injected into the sample for analysis (diluted to within the standard curve range according to the sample concentration). The samples were operated in parallel. A calibration curve (0.5–200 μg/mL) was prepared using an AQ standard on the same day as the sample analysis. The samples and standards were analyzed using a liquid chromatography (HPLC)/UV detector (Agilent 1260, Agilent Technologies, Inc., Santa Clara, CA, USA) and an Agilent Poroshell 120 EC-C18 (4.6 × 150 mm, 4 µm) column. The conditions for the liquid phase analysis were as follows: isocratic elution, mobile phases of ultrapure water (MPA) and acetonitrile (MPB), elution ratio MPA:MPB = 30:70 (*v*:*v*), flow rate 0.8 mL/min, column temperature: 30 °C, injection volume: 10 μL, total run time: 6.5 min. Chromatogram acquisition and peak integration of the analytes were performed by using the software Open Lab (Agilent), which automatically integrates the target chromatographic peaks after optimization of the integration parameters. The peak areas of AQ in the standard samples were recorded at different concentrations of AQ, and a linear regression was performed with the concentration of AQ as the horizontal coordinate and the peak area as the vertical coordinate. The resulting regression equation (Y = a + bX) was the standard curve. The concentration of AQ in μg/mL was back-calculated from this standard curve.

#### 2.4.2. Clinical Observations, Body Weight, and Food Consumption

All the animals were observed daily for abnormalities and mortality. Body weight was measured twice a week for the first four weeks and then weekly, and food consumption was recorded twice weekly.

#### 2.4.3. Hematology and Clinical Biochemistry

Hematology and clinical biochemistry were performed at the end of administration for the main group and the end of the recovery period for the satellite group. The rats were fasted for 16 h prior to blood collection. On the day of blood collection, the rats were anesthetized with pentobarbital sodium, and blood samples were collected from the abdominal aorta. Hematology indexes, including white blood cells (WBCs) and their classifications, such as lymphocytes (LYMPHs), neutrophils (NEUTs), eosinophils (EOs), monocyte (MO), and basophils (BASO), erythrocyte (RBC) count, hemoglobin (HGB) concentration, hematocrit (HCT) level, platelet (PLT) count, mean corpuscular volume (MCV), mean corpuscular hemoglobin (MCH), mean corpuscular hemoglobin concentration (MCHC), and reticulocyte (RET) count, were measured with a 2000i Hematology Analyzer (Sysmex, Tokyo, Japan). Prothrombin times (PTs) and activated partial thromboplastin times (APTTs) were assessed with an automated coagulation analyzer (Sysmex CA-510, Tokyo, Japan).

For serum chemistry analyses, blood samples were placed in a serum separator tube (gel barrier) and centrifuged. Serum chemistry parameters, including levels of alanine aminotransferase (ALT), aspartate aminotransferase (AST), total protein (TP), albumin (ALB), glucose (GLU), sorbitol dehydrogenase (SDH), blood urea nitrogen (BUN), creatinine (CRE), total cholesterol (CHO), alkaline phosphates (ALPs), triglycerides (TGs), lactate dehydrogenase (LDH), total bilirubin (T-Bil), high-density lipoprotein cholesterol (HDL-C), low-density lipoprotein cholesterol (LDL-C), cholinesterase (CHE), N-acetyl-β-D-glucosaminidase (NAG), Na^+^, K^+^, Cl^−^, and Ca^2+^, were determined with an Automated Biochemistry Analyzer (Beckman AU680, Pasadena, CA, USA).γ-GT isoenzyme (GGT) levels were measured using an enzyme-linked immunosorbent assay.

#### 2.4.4. Urinalysis

Urinalysis was performed on all rats at the end of the administration and the end of the recovery period. Urine samples were collected in stainless-steel metabolism cages over an approximately 20 h period. pH, glucose (GLU), protein (PRO), occult blood (BLD), specific gravity (SG), bilirubin (BIL), urobilinogen (URO), ketone body (KET), and WBCs were determined with a urine analyzer (Urist-H800, Changchun, China).

#### 2.4.5. Hormones

At the end of the administration or recovery period, samples were obtained from each animal, and serum total thyroxine (T4), triiodothyronine (T3), and thyroid-stimulating hormone (TSH) levels were measured. These hormones were measured by enzyme-linked immunosorbent assay.

#### 2.4.6. Sperm Motility and Vaginal Cytology Evaluations

Vaginal cytology and sperm viability were performed on male and female rats exposed to 174.08 mg/kg BW AQ at the end of the dosing and recovery periods, respectively. The vaginas of the female rats were gently flushed with saline, and the flushed saline was then applied to slides and examined microscopically to determine the stage of the estrous cycle (i.e., diestrus, proestrus, estrus, or metestrus). Sperm counts and motility were assessed in the males. One side of the epididymis was isolated from the male rats. The epididymis was weighed, and incisions were made. Sperm exiting the incisions were allowed to disperse in a buffer on the slides, and five areas of each slide were counted for active and inactive sperm counts using an automated sperm analyzer (Hamilton TOX IVOS II, Beverly, MA, USA).

#### 2.4.7. Pathology

At the end of the treatment or recovery period, the surviving animals were sacrificed by exsanguination. A full and detailed gross necropsy was performed on all the animals. Organs, including the liver, kidneys, adrenals, heart, lungs, spleen, testes, thymus, epididymis, uterus, ovaries, thyroid parathyroid glands, prostate seminal vesicles, and brain, were dissected and weighed quickly. The relative organ weight was expressed as a percentage of the final individual BW.

The following organs and tissues were examined under a microscope: brain, pituitary, eyes, spinal cord (cervical, thoracic, lumbar), thymus, thyroid, parathyroid, heart, lung, liver, spleen, kidneys, adrenals, stomach, duodenum, pancreas, jejunum, ileum, colon, rectum, lymphonodi mesosteniales, testes, epididymides, uterus, ovaries, urinary bladder, prostate, breast, skin, muscles, and sternum. All these samples were routinely processed for embedding in paraffin, ectioned, and then stained with hematoxylin and eosin. Kidney tissue from male rats in all dose groups was then re-sampled, paraffin embedded, sectioned, immunohistochemically stained (alpha-2u globulin), and graded for evaluation. Microscopic evaluation was performed by a pathologist. The histomorphological findings were graded on a scale of 1 to 5, with 1 indicating nominal (in the study condition, the tissue is considered normal considering the age, sex and strain of the animal), 2 minimal (changes just outside the normal range), 3 mild ( Lesions can be observed, but are not severe), 4 moderate (lesions are obvious and likely to be more severe), and 5 severe (the lesion is very severe and has taken over the entire tissue and organs) [17].

### 2.5. Statistical Analysis

The mean standard deviation of response variable values was calculated for each gender. All quantitative variables collected, such as body weight and food consumption, clinical chemistry, and tissue organs, were tested for normality and variance homogeneity. Multiple data sets between dose and control groups were analyzed using one-way analysis of variance (ANOVA) tests and Dunnett *t*-tests for multiple comparisons, as well as *t*-tests for satellite control and satellite high-dose groups, when variances were homogeneous. When the data were found to be heterogeneous, a suitable transformation was performed. The fisher exact probability method was used for pathological examination statistics, and the Kruskal–Wallis rank sum test was used for urine parameters, lesion extent, and other nonparametric data. All analyses and comparisons were performed at a 5% (*p* = 0.05) level.

## 3. Results

### 3.1. Oral Acute Toxicity Study

The rats repeatedly exposed to AQ at a dose of 2000 mg/kg BW twice were observed with no obvious abnormalities during the 14-day observation period. BW gain and gross necropsy showed no related changes in either sex.

### 3.2. Subchronic Toxicity in Rats

#### 3.2.1. Formulation, Analysis, and Storage

The initial doses of the 1.36 mg/kg BW and 174.08 mg/kg BW groups were 96.9% and 99.5% of the theoretical concentrations, respectively, before the start of the test, with deviations of 3.2% and 0.4%, respectively, after 72 h at room temperature. The results showed that the stability of AQ in 1% sodium carboxymethylcellulose solution at room temperature for at least 72 h was good. This also indicated that the AQ formulation can be stored at room temperature and prepared every 3 days.

At a dose of 1.36 mg/kg bw (0.136 mg/mL), the relative standard deviation (RSD) of AQ concentrations in the upper, middle, and lower samples was 1.4%, with a mean assay concentration of 0.129 mg/mL. At 5.44 mg/kg bw (0.544 mg/mL), the RSD was 2.9%, and the mean assay concentration in the upper, middle, and lower samples was 0.559 mg/mL. The RSD of AQ concentrations in the upper, middle, and lower samples was 1.2%, the mean assay concentrations were 2.172 mg/mL and 174.08 mg/kg bw (17.408 mg/mL), and the RSD for AQ in the upper, middle, and lower samples was 1.5%, with a mean assay concentration of 18.559 mg/mL; these values were 94.9%, 102.8%, 99.8%, and 106.6% of their theoretical concentrations, respectively (Figure 1).

#### 3.2.2. Clinical Observations, Body Weight, and Food Consumption

During the study, one male animal at 174.08 mg/kg body weight died on day 65 with no clinical signs prior to death. The pathological examination revealed that the lung congestion and edema were accompanied by cellulose-like substance exudation, but the cause of death could not be determined. The surviving rats in the other dose groups were observed to have no obvious abnormalities.

No significant differences in body weight were produced between male and female rats at each dose compared to the control group. However, the male rats at 174.08 mg/kg bw had reduced total food consumption for 13 weeks. During the recovery period, the weight and food consumption of the rats in the other dose groups were not significantly different from those in the satellite control group (Figure 2 and Figure 3).

#### 3.2.3. Hematology and Clinical Biochemistry

At the end of administration, compared to those in the control group, the PLT count and HCT% increased in the female 1.36 mg/kg BW rats; the PLT count, MCV, MCH, and RET% increased in the 5.44 mg/kg BW rats; the LY%, PLT count, MCV, MCH, and RET% increased in the 21.76 mg/kg BW rats, and the NEUT%, RBCs, and MCHC decreased; in the 174.08 mg/kg BW rats, the MCV, MCH, and RET% increased, and the EO%, RBCs, MCHC, and PT decreased; the MCV and RET% increased in the male 21.76 mg/kg BW rats; and the MCV increased and MCHC decreased in the 174.08 mg/kg BW rats. During the recovery period, the RET% decreased in the male satellite 174.08 mg/kg BW rats compared to the satellite negative controls. The results are detailed in Table 1.

At the end of the study, compared with that in the negative control group, a decreased SDH dose-related decline in the female rats was observed in the 5.44, 21.76, and 174.08 mg/kg BW groups. The 174.08 mg/kg BW rats’ levels of TP, ALB, CHOL, TG, HDL-C, LDL-C, and Na+ increased, and the levels of ALP, CHE, and SDH decreased. The GLU levels of the male rats in the 5.44 and 21.76 mg/kg BW groups were higher, the NAG level of the rats in the 21.76 mg/kg BW group increased, and the CHOL, HDL-C, LDL-C, and NAG levels increased in the 174.08 mg/kg BW group, while the levels of AST and LDH decreased. During the recovery period of the study, compared with the satellite negative control group, the female satellite 174.08 mg/kg BW group had higher K+ and Cl- and decreased ALT levels; the male satellite 174.08 mg/kg BW group’s levels of TP and TBIL decreased. No abnormalities were observed in the other serum biochemical indexes of each dose group at the middle point of the test, at the end of the test, or during the recovery period. The results are detailed in Table 2.

#### 3.2.4. Urinalysis

Urinalysis parameters, including pH, NIT, GLU, SG, PRO, BLD, BIL, UBG, and KET, were within normal limits. No significant changes were observed in the dose groups.

#### 3.2.5. Hormones

At the end of the study, T4 decreased in the 5.44 and 174.08 mg/kg BW female groups, and T3 decreased in the 21.76 and 174.08 mg/kg BW male groups. The results are detailed in Table 3.

#### 3.2.6. Sperm Motility and Vaginal Cytology Evaluations

No significant changes were found in the dose groups.

#### 3.2.7. Organ Weights

Compared to the values in the control group, the absolute and relative organ weights of the liver, kidney, and spleen were increased in the rats in the 21.76 and 174.08 mg/kg BW groups, and the thyroid weight was decreased in the female rats in the 174.08 mg/kg BW group. The liver weight and liver-to-body ratio were higher in the male rats in the 21.76 mg/kg BW group, and the 174.08 mg/kg BW group rats had higher liver and liver-to-body ratios and higher kidney and kidney-to-body ratios.

No significant differences were observed in organs between the satellite groups at the end of the recovery period. The results are detailed in Table 4.

#### 3.2.8. Pathology

Following 13 weeks of exposure, there was minimal centrilobular hypertrophy of the liver in the 174.08 mg/kg BW female rats, and hyaline droplet accumulation of the kidney increased in incidence in the female (≥5.44 mg/kg BW) and male (≥1.36 mg/kg BW) rats (Figure 4, Table 5). However, no significant difference was observed between the liver and kidney pathology of the 174.08 mg/kg BW group and the control group after the recovery period.

The hyaline droplet accumulation of the kidney of the rats was then further graded and evaluated, with the degree of lesions starting at 5.44 mg/kg BW significantly higher in the females and males than in the controls at the end of the administration. No hyaline droplet accumulation of the kidney was observed in the females at 1.36 mg/kg BW, and hyaline droplet accumulation of the kidney was observed in the males at 1.36 mg/kg BW, but there was no significant difference when compared to the control group (Table 6).

Kidney tissues from the male rats of all the dose groups were resampled, paraffin-embedded, sectioned, immunohistochemically stained for α-2u globulin, and graded for evaluation. The results showed that the kidney tissues of the male rats in the control and all the dose groups were positive for α-2u globulin immunohistochemical staining, and the grading statistics of immunohistochemistry of the animals in each group showed no significant difference between the groups (Figure 5, Table 7).

## 4. Discussion

To provide safety evaluation data, the acute and subchronic oral toxicity of SD rats were evaluated.

In an acute oral toxicity test, six rats (three rats per step, two steps) were given orally 2000 mg/kg anthraquinone. No obvious toxic symptoms or death were found in the 14 days. According to the OECD guideline 423 appendix 2D grading standard, the test substance belongs to five or no categories in the Globally Harmonized Classification System (GHS) classification, with LD_50_ ≥ 5000 mg/kg BW. The results of acute oral toxicity test are consistent with the previous literature.

During the subchronic toxicity test, the AQ formulation was analyzed three times, and each time, the upper, middle, and lower AQ formulation concentrations and the mean concentration values were determined to be between 85% and 115% of the theoretical concentration, indicating that the concentrations of the AQ formulations met the requirements of the test design. The results of the AQ subchronic toxicity test showed a decrease in food consumption in the male rats in the 174.08 mg/kg BW group, but there was no significant difference in body weight in males, so the reduction in food consumption alone was not toxicologically significant.

Combining the hematological data from the end of the study and the recovery period, there were some parameters (RBC, HCT%, MCV, MCH, MCHC) that were somewhat different compared to the control group, but the increase or decrease was within 10% of the control group mean, and the measurements were within the normal range for this laboratory, indicating no toxicological significance. The PLT count increased significantly (*p* < 0.05 or *p* < 0.01) in the female 1.36, 5.44, and 21.76 mg/kg BW groups, but the difference between the PLT count at 174.08 mg/kg BW and the control group was not significant (*p* > 0.05), indicating that the increase in the PLT count was an adaptive response. Some of the indicators were not abnormal at the end of the dosing period but rose or fell during the recovery period, and these changes were not thought to be due to AQ. At the end of the test, the RET% of the female rats increased from a dose of 5.44 mg/kg BW with a significant difference (*p* < 0.01) in a dose–response relationship. During the recovery period, the RET% values of the rats did not differ from those of the control group, and the assay values were outside the standard range for this laboratory, indicating that 5.44 mg/kg BW AQ causes anemia in female rats, but after the recovery period, rats can recover from the effects of AQ on the RET%.

Combined with the clinical biochemical data at the end of the study and the recovery period of the test, the levels of CHOL, TG, HDL-C, and LDL-C increased, and those of ALP and CHE decreased in the female 174.08 mg/kg BW rats at the end of the test, the level of NAG increased in the male 21.76 mg/kg BW group rats and those of CHOL, HDL-C, LDL-C, and NAG increased in the 174.08 mg/kg BW group rats, suggesting that AQ has certain effects on the liver and kidney of rats, but after the recovery period, rats can recover from the effects of AQ on the liver and kidney. These changes are consistent with the lesions in the liver and kidney revealed in the pathological examination. The liver, kidney, and organ coefficients of the 174.08 mg/kg BW male and female rats showed an increase, and pathological findings showed a small amount of inflammatory cell infiltration in the female rats in the174.08 mg/kg BW group. Slight tubular eosinophilic glassy droplets were observed in the kidneys in the female rats at ≥5.44 mg/kg BW and in the male rats at all doses.

In the hormone examination, the level of T4 decreased in the 174.08 mg/kg BW group in the female rats, and that of T3 decreased in the 21.76 mg/kg BW and 174.08 mg/kg BW groups in the male rats. In combination with the high CHOL, HDL-C and LDL-C levels in the 174.08 mg/kg BW group in both the female and male rats at the end of the study, the reduced thyroid gland in the 174.08 mg/kg BW female rats in terms of organ weight suggests that 174.08 mg/kg BW AQ may have some effect on hypothyroidism in female rats. However, after a recovery period, rats can recover from the effects of AQ on hypothyroidism.

Pathological immunohistochemistry showed that the eosinophilic glassy droplets observed in the male rats at all doses were not caused by the aggregation of alpha-2u globulin, and that a toxic reaction due to the test substance could not be excluded. In combination with HE staining pathology, the test caused mainly slight hepatocellular hypertrophy around the central vein of the hepatic lobule in the 174.08 mg/kg BW group of female rats and eosinophilic glassy droplet pathology in the kidneys of the ≥5.44 mg/kg male and female rats, with statistically significant differences. The pathological histological changes noted at the end of the AQ exposure period could not be observed at the end of the recovery period.

This study’s findings are more consistent with previous findings in the literature, such as increased liver and kidney weights, mild centrilobular hypertrophy in the liver, minimal hyaline droplets in the kidney, and female anemia. In this study, female rats had more anemia, hypothyroidism, and liver pathological changes, but male rats had more hyalinized droplet changes in the kidney, and anthraquinone >5.44 mg/kg BW increased the degree of hyalinized droplet changes in the kidney. Adult male rats have a specific protein (a-2u globulin) that accumulates and deposits in the tubular epithelium and can be seen as hyaline droplets using hematoxylin-eosin (HE) staining, which resembles the pathological changes seen in our assay. As a result, we added an immunohistochemical test for a-2u globulin to see if male rats were more sensitive to this protein. Anthraquinone did not induce a-2u globulin in male rats in this experiment, despite NTP 14-week and carcinogenicity tests showing an increase in a2u globulin in male rat kidneys at anthraquinone levels greater than 3750 ppm (275 mg/kg). Simultaneously, we reviewed the literature on hyalinized droplets, and the data revealed that hyalinized droplets in the kidney are an accessory mechanism in the metabolic process of absorptive protein disposal by the kidney and should be explained by quantitative rather than qualitative factors [18,19]. Renal carcinogens can increase renal hyaline droplets in male rats [20], and it has also been reported that hyalinized droplets changes in the kidney are some column changes associated with a-2u globulin accumulation in the kidney [21], but the relationship between renal carcinogens and renal hyaline droplets is unknown.

## 5. Conclusions

Collectively, the LD50 of oral AQ administered to SD rats was estimated to be >5000 mg/kg BW and classified into 5 or no categories in the GHS. In a 90-day oral subchronic test in SD rats, the NOAEL for AQ was considered to be 1.36 mg/kg BW. When AQ was gavaged at 5.44 mg/kg BW or higher, symptoms such as anemia, hepatotoxicity and nephrotoxicity were observed and were more pronounced in females.

## Figures and Tables

**Figure 1 ijerph-19-10413-f001:**
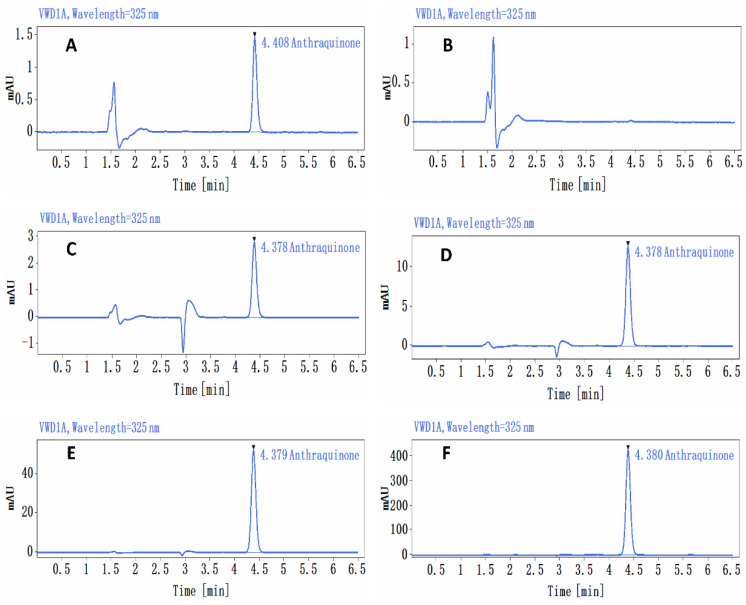
Chromatogram of the analysis of the anthraquinone preparations. (**A**) Standard curve sample (0.5 μg/mL) chromatogram, (**B**) control solvent 1% CMC-Na chromatogram, (**C**) anthraquinone preparation (0.136 mg/mL) chromatogram, (**D**) anthraquinone preparation (0.544 mg/mL) chromatogram, (**E**) anthraquinone preparation (2.176 mg/mL) chromatogram, (**F**) anthraquinone preparation (17.408 mg/mL) chromatogram.

**Figure 2 ijerph-19-10413-f002:**
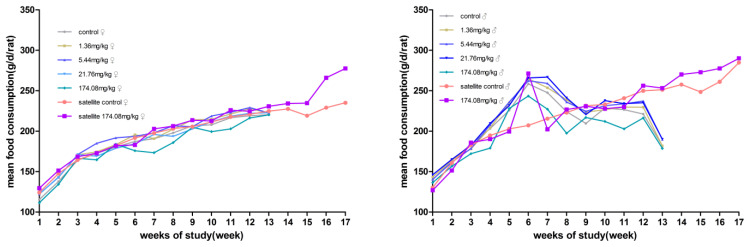
Average body weight of rats orally administered with AQ for 90 days. (**Left**) female; (**Right**) male.

**Figure 3 ijerph-19-10413-f003:**
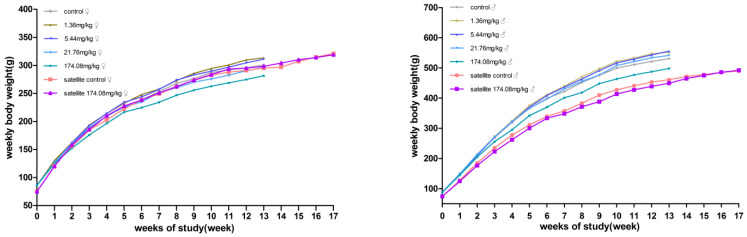
Average daily food consumption of the rats orally administered AQ for 90 days. (**Left**) female; (**Right**) male.

**Figure 4 ijerph-19-10413-f004:**
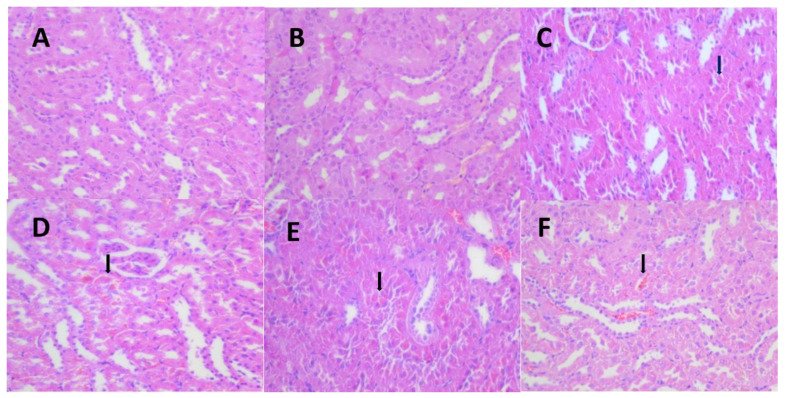
**Representative images of the pathological changes in the Hyaline droplet (arrow) of the kidney tissue in each dose group and the control group at 13 weeks.** (**A**) (HE 40 × 10, control group, normal kidney), (**B**) (HE 40 × 10, 1.36 mg/kg group, normal kidney), (**C**) (HE 40 × 10, 1.36 mg/kg group, minimalhyaline droplets in the kidney), (**D**) (HE 40 × 10, 5.44 mg/kg group, mild hyaline droplets in the kidney), (**E**) (HE 40 × 10, 21.76 mg/kg group, mild hyaline droplets in the kidney), (**F**) (HE 40 × 10, 174.08 mg/kg group, mild hyaline droplets in the kidney).

**Figure 5 ijerph-19-10413-f005:**
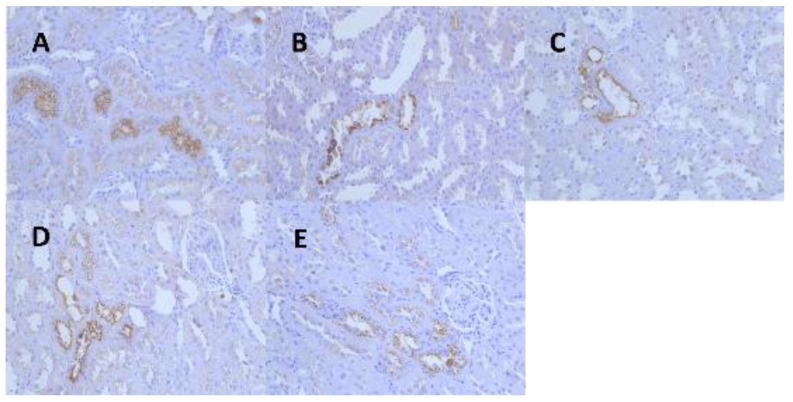
Representative images of renal α-2u globulin in each dose group and control group at 13 weeks. (**A**) (IHC 40 × 10, control group), (**B**) (IHC 40 × 10, 1.36 mg/kg group), (**C**) (IHC 40 × 10, 5.44 mg/kg group), (**D**) (IHC 40 × 10, 21.76 mg/kg group), (**E**) (IHC 40 × 10, 174.08 mg/kg group).

**Table 1 ijerph-19-10413-t001:** Hematological values in SD rats orally administered AQ for 90 days.

Organ	Units	End of Treatment Period (mg/kg)										End of Recovery Period (mg/kg)			
		0.00		1.36		5.44		21.76		174.08		0.0		174.08	
		♀ n = 10	♂ n = 10	♀ n = 10	♂ n = 10	♀ n = 10	♂ n = 10	♀ n = 10	♂ n = 10	♀ n = 10	♂ n = 9	♀ n = 5	♂ n = 5	♀ n = 5	♂ n = 5
WBC	10^9^/L	4.2 ± 1.2	6.7 ± 1.6	4.8 ± 1.4	5.9 ± 1.7	4.9 ± 0.9	6.1 ± 1.5	5.1 ± 1.2	7.6 ± 1.4	4.1 ± 1.2	6.7 ± 1.3	3.4 ± 0.7	7.3 ± 1.7	3.4 ± 0.7	6.0 ± 0.7
LY	%	63.0 ± 3.4	62.5 ± 5.0	65.7 ± 4.2	63.4 ± 5.6	68.6 ± 5.5	61.1 ± 5.6	69.8 ± 4.3 *	66.1 ± 6.5	67.8 ± 7.2	61.8 ± 5.2	58.8 ± 13.6	49.1 ± 9.7	64.0 ± 8.7	51.6 ± 5.5
MO	%	1.5 ± 0.4	1.8 ± 0.4	1.5 ± 0.4	1.6 ± 0.4	1.2 ± 0.4	2.0 ± 0.4	1.2 ± 0.3	1.5 ± 0.5	1.6 ± 0.7	2.0 ± 0.8	1.3 ± 0.6	2.5 ± 1.5	1.4 ± 0.3	2.7 ± 0.7
NEUT	%	33.3 ± 3.3	33.9 ± 4.9	30.7 ± 3.8	32.6 ± 5.5	28.6 ± 5.4	35.0 ± 5.8	27.4 ± 4.0 *	30.4 ± 5.9	29.3 ± 7.1	34.1 ± 4.8	37.1 ± 13.4	46.5 ± 9.3	32.0 ± 7.8	43.7 ± 5.4
EO	%	2.3 ± 0.8	1.8 ± 0.3	2.2 ± 0.7	2.4 ± 1.2	1.6 ± 0.5	2.0 ± 0.5	1.7 ± 0.7	2.1 ± 0.5	1.4 ± 0.5 *	2.1 ± 0.7	2.8 ± 0.6	2.0 ± 0.3	2.6 ± 1.0	2.0 ± 0.3
BASO	%	0.0 ± 0.0	0.0 ± 0.0	0.0 ± 0.0	0.0 ± 0.0	0.0 ± 0.0	0.0 ± 0.0	0.0 ± 0.0	0.0 ± 0.0	0.0 ± 0.0	0.0 ± 0.0	0.0 ± 0.0	0.0 ± 0.0	0.0 ± 0.0	0.0 ± 0.0
RBC	10^12^/L	7.9 ± 0.5	8.4 ± 0.4	8.1 ± 0.2	8.3 ± 0.4	7.6 ± 0.3	8.2 ± 0.6	7.4 ± 0.4 *	8.0 ± 0.4	7.4 ± 0.5 *	8.1 ± 0.3	8.2 ± 0.6	8.4 ± 0.4	8.2 ± 0.5	8.4 ± 0.3
HGB	g/L	147.0 ± 8.0	150.1 ± 4.6	153.9 ± 6.5	147.8 ± 7.4	147.5 ± 5.7	146.8 ± 7.9	144.1 ± 6.7	146.5 ± 6.5	143.3 ± 6.8	147.9 ± 6.7	150.0 ± 7.3	148.4 ± 3.5	152.4 ± 6.6	149.2 ± 2.5
PLT	10^9^/L	1001.0 ± 151.9	1150.2 ± 126.8	1224.6 ± 168.5 **	1133.2 ± 171.0	1218.2 ± 102.7 **	1170.1 ± 87.9	1204.4 ± 148.3 *	1241.1 ± 110.0	1142.0 ± 161.0	1121.1 ± 90.8	1154.6 ± 193.7	1148.6 ± 167.3	1127.0 ± 329.4	1289.2 ± 127.6
HCT	%	42.3 ± 2.2	43.0 ± 1.1	44.4 ± 1.8 *	42.4 ± 2.0	43.1 ± 1.2	42.6 ± 2.1	42.7 ± 1.8	42.7 ± 2.0	43.1 ± 2.0	43.2 ± 1.7	42.5 ± 1.8	42.6. ± 1.6	43.6 ± 1.6	42.3 ± 1.7
MCV	f1	53.4 ± 1.7	51.4 ± 1.1	54.9 ± 1.6	51.1 ± 1.5	56.5 ± 1.8 **	52.3 ± 1.9	57.8 ± 1.7 **	53.3 ± 1.7 *	58.3 ± 1.9 **	53.7 ± 1.7 *	52.2 ± 3.1	50.5 ± 1.4	53.1 ± 1.9	50.1 ± 1.0
MCH	pg	18.5 ± 0.4	18.0 ± 0.5	19.0 ± 0.5	17.8 ± 0.5	19.3 ± 0.5 **	18.1 ± 0.5	19.5 ± 0.5 **	18.3 ± 0.6	19.4 ± 0.6 **	18.4 ± 0.4	18.4 ± 0.6	17.6 ± 0.4	18.5 ± 0.5	17.7 ± 0.3
MCHC	g/L	347.1 ± 4.8	349.6 ± 5.4	346.5 ± 4.5	348.8 ± 5.0	342.2 ± 5.6	344.9 ± 6.9	337.0 ± 3.6 **	343.1 ± 4.9	332.4 ± 3.1 **	342.4 ± 7.1 *	353.0 ± 12.7	348.8 ± 6.9	349.6 ± 5.2	353.2 ± 8.8
RET	%	2.0 ± 0.2	2.3 ± 0.3	2.2 ± 0.5	2.4 ± 0.6	2.9 ± 0.5 **	2.3 ± 0.3	3.5 ± 0.7 **	2.9 ± 0.2 *	4.1 ± 0.9 **	2.6 ± 0.8	1.6 ± 0.4	2.3 ± 0.2	1.8 ± 0.5	1.9 ± 0.2 #
APTT	(S)	9.4 ± 2.9	7.0 ± 1.4	8.3 ± 1.8	7.6 ± 4.6	8.7 ± 1.8	8.6 ± 3.1	8.5 ± 1.5	7.4 ± 1.5	8.6 ± 1.7	8.1 ± 1.7	11.4 ± 8.6	6.6 ± 1.0	6.9 ± 1.0	6.3 ± 1.2
PT	(S)	13.6 ± 0.8	12.8 ± 0.5	13.1 ± 0.5	12.9 ± 0.5	13.5 ± 0.4	12.7 ± 0.9	13.2 ± 0.5	12.9 ± 0.5	13.0 ± 0.4 *	12.4 ± 0.5	12.1 ± 0.5	13.2 ± 2.3	12.2 ± 0.3	11.6 ± 1.6

* Represents a significant difference at the *P*< 0.05 level compared with the control group. ** Represents a significant difference at the *p* < 0.01 level compared with the control group. # Represents a significant difference at the *p* < 0.05 level compared with the satellite control group.

**Table 2 ijerph-19-10413-t002:** Blood chemistry values in SD rats orally administered AQ for 90 days.

Organ	Units	End of Treatment Period (mg/kg)										End of Recovery Period (mg/kg)			
		0.00		1.36		5.44		21.76		174.08		0.0		174.08	
		♀ n = 10	♂ n = 10	♀ n = 10	♂ n = 10	♀ n = 10	♂ n = 10	♀ n = 10	♂ n = 10	♀ n = 10	♂ n = 9	♀ n = 5	♂ n = 5	♀ n = 5	♂ n = 5
ALT	U/L	39.6 ± 6.2	39.3 ± 6.1	39.7 ± 9.3	38.3 ± 8.3	41.0 ± 5.5	37.7 ± 7.4	39.1 ± 7.5	34.6 ± 2.6	36.5 ± 5.1	36.1 ± 6.6	41.0 ± 3.9	50.6 ± 13.5	30.8 ± 5.6 #	45.0 ± 18.6
AST	U/L	115.9 ± 25.6	93.1 ± 10.6	114.3 ± 19.5	96.9 ± 26.4	131.6 ± 18.1	88.9 ± 8.0	103.6 ± 16.5	88.5 ± 7.3	100.9 ± 16.4	76.7 ± 10.8 *	112.6 ± 13.7	123.6 ± 22.9	98.2 ± 21.2	99.4 ± 28.0
ALP	IU/L	47.7 ± 8.5	72.4 ± 12.9	47.5 ± 10.9	69.5 ± 6.1	48.7 ± 10.5	79.6 ± 8.7	39.6 ± 9.6	70.6 ± 5.1	36.3 ± 10.4 *	68.0 ± 16.5	43.6 ± 5.8	73.6 ± 17.7	42.4 ± 9.0	63.8 ± 17.8
GGT	ng/mL	3.6 ± 0.4	4.0 ± 0.4	4.0 ± 0.3	3.6 ± 0.2	3.8 ± 0.2	4.1 ± 0.4	3.6 ± 0.4	4.0 ± 0.6	3.7 ± 0.6	4.2 ± 0.6	2.7 ± 0.2	2.6 ± 0.1	3.0 ± 0.7	2.8 ± 0.4
LDH	IU/L	801.7 ± 289.5	714.1 ± 146.6	922.2 ± 194.8	664.1 ± 259.9	1067.9 ± 209.8	653.4 ± 200.5	806.6 ± 235.1	689.0 ± 117.3	807.0 ± 302.3	442.6 ± 198.5 *	687.6 ± 115.1	399.8 ± 268.2	703.0 ± 162.2	467.0 ± 222.5
CHE	IU/L	2037.3 ± 224.8	1466.7 ± 75.9	1986.7 ± 278.6	1501.5 ± 64.4	1959.4 ± 156.8	1538.4 ± 270.3	1971.6 ± 206.6	1499.3 ± 110.9	1741.4 ± 193.7 *	1451.0 ± 175.7	2108.0 ± 286.3	1460.2 ± 73.3	2143.6 ± 194.9	1430.6 ± 34.4
NAG	ng/mL	19.9 ± 2.4	16.1 ± 2.7	18.1 ± 3.9	13.3 ± 1.7	16.4 ± 4.3	18.6 ± 2.6	17.9 ± 2.1	21.1 ± 2.5 **	16.3 ± 3.4	21.4 ± 3.2 **	13.6 ± 1.6	14.7 ± 1.2	17.9 ± 7.2	12.9 ± 1.4
SDH	ng/mL	2.5 ± 0.2	1.9 ± 0.5	2.1 ± 0.3	1.6 ± 0.3	2.0 ± 0.5 *	2.3 ± 0.2	2.0 ± 0.4*	2.1 ± 0.7	1.6 ± 0.4 **	2.0 ± 0.7	1.0 ± 0.1	1.0 ± 0.1	0.9 ± 0.2	0.8 ± 0.3
TP	g/L	57.9 ± 4.3	52.7 ± 1.5	58.6 ± 2.6	54.2 ± 3.4	57.3 ± 1.5	53.6 ± 2.3	58.6 ± 2.2	54.5 ± 2.3	63.9 ± 3.8 **	55.5 ± 3.2	61.2 ± 1.2	58.6 ± 2.8	60.4 ± 3.2	54.8 ± 1.6 #
ALB	g/L	33.0 ± 2.2	28.1 ± 1.2	33.0 ± 1.6	28.0 ± 1.2	32.0 ± 0.8	28.5 ± 0.8	33.0 ± 1.4	28.6 ± 0.8	35.1 ± 2.1 *	28.9 ± 1.6	32.8 ± 1.3	30.1 ± 1.4	32.7 ± 2.5	28.8 ± 0.3
CHOL	mmol/L	1.9 ± 0.4	2.1 ± 0.2	2.1 ± 0.5	2.3 ± 0.3	2.0 ± 0.6	2.3 ± 0.5	2.3 ± 0.3	2.4 ± 0.3	3.2 ± 0.5 **	2.9 ± 0.8 **	2.8 ± 0.4	2.3 ± 0.2	2.5 ± 0.4	1.9 ± 0.4
TG	mmol/L	0.6 ± 0.1	0.5 ± 0.2	0.7 ± 0.1	0.6 ± 0.2	0.6 ± 0.1	0.7 ± 0.2	0.6 ± 0.1	0.8 ± 0.2	0.7 ± 0.1 *	0.9 ± 0.5	0.6 ± 0.1	0.7 ± 0.2	0.7 ± 0.1	0.6 ± 0.2
HDL-C	mmol/L	1.8 ± 0.3	1.6 ± 0.2	1.8 ± 0.3	1.7 ± 0.2	1.8 ± 0.4	1.8 ± 0.3	2.0 ± 0.2	1.8 ± 0.1	2.6 ± 0.3 **	2.1 ± 0.5 **	2.1 ± 0.3	1.7 ± 0.1	2.0 ± 0.3	1.5 ± 0.2
LDL-C	mmol/L	1.0 ± 0.2	1.1 ± 0.1	1.1 ± 0.3	1.2 ± 0.2	1.1 ± 0.3	1.2 ± 0.3	1.2 ± 0.2	1.3 ± 0.2	1.8 ± 0.3 **	1.5 ± 0.5 **	1.4 ± 0.2	1.2 ± 0.2	1.3 ± 0.2	0.9 ± 0.2
GLU	mmol/L	6.7 ± 0.7	5.9 ± 0.7	7.2 ± 1.2	7.4 ± 0.8 *	6.0 ± 0.9	7.3 ± 1.0 *	5.9 ± 0.9	6.2 ± 1.3	5.6 ± 1.3	5.6 ± 1.6	6.9 ± 1.0	6.6 ± 1.0	6.1 ± 1.4	7.0 ± 1.0
BUN	mmol/L	3.6 ± 0.5	3.6 ± 0.2	3.4 ± 0.8	3.2 ± 0.4	4.0 ± 0.8	3.2 ± 0.4	3.7 ± 0.5	3.4 ± 0.8	3.1 ± 0.8	3.4 ± 0.6	3.6 ± 0.5	4.0 ± 0.8	3.8 ± 0.1	3.3 ± 0.4
CREA	umol/L	74.9 ± 7.3	92.7 ± 10.3	77.3 ± 10.8	91.5 ± 15.9	80.1 ± 7.4	91.4 ± 8.6	71.4 ± 2.9	89.7 ± 6.7	72.5 ± 5.6	82.0 ± 9.5	87.0 ± 9.9	122.0 ± 39.7	80.0 ± 8.7	80.7 ± 7.3
TBIL	umol/L	2.9 ± 0.5	2.3 ± 0.4	2.4 ± 0.5	2.4 ± 0.4	2.4 ± 0.2	2.3 ± 0.3	2.4 ± 0.5	2.2 ± 0.3	2.6 ± 0.5	2.0 ± 0.3	2.8 ± 0.3	2.6 ± 0.3	2.6 ± 0.4	2.2 ± 0.1#
K^+^	mmol/L	5.3 ± 0.4	4.9 ± 0.3	5.2 ± 0.6	4.5 ± 0.4	5.4 ± 0.3	4.6 ± 0.4	5.4 ± 0.3	4.9 ± 0.4	5.1 ± 0.2	4.7 ± 0.7	4.6 ± 0.1	5.3 ± 0.3	5.0 ± 0.2 #	5.2 ± 0.4
Na^+^	mmol/L	141.4 ± 2.0	138.0 ± 1.4	142.0 ± 2.2	138.0 ± 1.2	141.9 ± 1.8	138.3 ± 1.4	142.2 ± 2.9	137.8 ± 1.4	144.4 ± 3.2 *	137.8 ± 1.8	139.6 ± 1.5	141.9 ± 2.0	140.4 ± 1.4	142.3 ± 0.9
Cl^-^	mmol/L	107.2 ± 2.3	103.2 ± 1.6	107.7 ± 3.1	101.8 ± 1.7	107.9 ± 2.0	102.1 ± 2.3	108.6 ± 2.5	102.2 ± 2.2	107.6 ± 1.7	102.1 ± 2.1	104.9 ± 1.1	105.2 ± 3.8	107.0 ± 1.4 #	108.5 ± 1.3
Ca^2+^	mmol/L	2.9 ± 0.1	2.8 ± 0.1	2.9 ± 0.1	2.8 ± 0.1	2.9 ± 0.1	2.8 ± 0.1	2.9 ± 0.1	2.8 ± 0.1	3.0 ± 0.1	2.9 ± 0.1	2.9 ± 0.0	3.0 ± 0.3	2.9 ± 0.1	2.9 ± 0.0

* Represents a significant difference at the *p* < 0.05 level compared with the control group. ** Represents a significant difference at the *p* < 0.01 level compared with the control group. # Represents a significant difference at the *p* < 0.05 level compared with the satellite control group.

**Table 3 ijerph-19-10413-t003:** Hormones in SD rats orally administered AQ for 90 days.

Organ	Units	End of Treatment Period (mg/kg)										End of Recovery Period (mg/kg)			
		0.00		1.36		5.44		21.76		174.08		0.0		174.08	
		♀ n = 10	♂ n = 10	♀ n = 10	♂ n = 10	♀ n = 10	♂ n = 10	♀ n = 10	♂ n = 10	♀ n = 10	♂ n = 9	♀ n = 5	♂ n = 5	♀ n = 5	♂ n = 5
T3	ng/mL	1.0 ± 0.2	1.1 ± 0.1	1.0 ± 0.1	1.0 ± 0.1	1.0 ± 0.1	1.1 ± 0.1	1.1 ± 0.1	0.9 ± 0.1 **	1.0 ± 0.1	0.9 ± 0.1 *	0.9 ± 0.1	0.9 ± 0.1	0.8 ± 0.1	0.8 ± 0.1
T4	ng/mL	27.8 ± 2.0	26.1 ± 1.3	27.9 ± 2.3	26.0 ± 2.0	24.8 ± 2.6 *	27.5 ± 3.8	27.9 ± 3.2	25.9 ± 2.9	23.7 ± 2.1 **	28.1 ± 3.141	10.521 ± 2.131	12.841 ± 3.2	14.3 ± 1.3	16.7 ± 1.2 #
TSH	mU/L	2.5 ± 0.5	2.5 ± 0.7	2.2 ± 0.5	2.1 ± 0.5	2.2 ± 0.3	2.7 ± 0.3	2.5 ± 0.5	2.1 ± 0.1	2.1 ± 0.4	2.2 ± 0.2	1.9 ± 0.4	2.1 ± 0.2	1.9 ± 0.3	2.1 ± 0.3

* Represents a significant difference at the *p* < 0.05 level compared with the control group. ** Represents a significant difference at the *p* < 0.01 level compared with the control group. # Represents a significant difference at the *p* < 0.05 level compared with the satellite control group.

**Table 4 ijerph-19-10413-t004:** Absolute and relative (to body weight) organ weights of SD rats orally administered AQ for 90 days.

Organ	Units	End of Treatment Period (mg/kg)										End of Recovery Period (mg/kg)			
		0.00		1.36		5.44		21.76		174.08		0.0		174.08	
		♀ n = 10	♂ n = 10	♀ n = 10	♂ n = 10	♀ n = 10	♂ n = 10	♀ n = 10	♂ n = 10	♀ n = 10	♂ n = 9	♀ n = 5	♂ n = 5	♀ n = 5	♂ n = 5
liver	g ^a^	7.1 ± 0.7	12.7 ± 1.4	7.4 ± 0.8	13.7 ± 1. 8	7.5 ± 0.9	14.3 ± 1.5	14.6 ± 0. 8	14.7 ± 1.9 *	9.7 ± 1.3 **	16.5 ± 1.6 **	7.3 ± 0.3	14.6 ± 0. 8	7.2 ± 0.5	15.0 ± 0.3
	% ^b^	2.537 ± 0.107	2.5 ± 0.1	2.5 ± 0.2	2.6 ± 0.2	2.6 ± 0.2	2.7 ± 0.1	3.2 ± 0.3	2.8 ± 0.2 **	3.7 ± 0.2 **	3.5 ± 0.3 **	2.4 ± 0.1	3.2 ± 0.3	2.4 ± 0.2	3.2 ± 0.1
kidney	g ^a^	1.9 ± 0.2	3.5 ± 0.5	1.8 ± 0.1	3.5 ± 0.6	1.9 ± 0.2	3.9 ± 0.3	3.2 ± 0.2	4.0 ± 0.3	2.0 ± 0.2	4.1 ± 0.7 *	1.8 ± 0.1	3.2 ± 0.2	1. 8 ± 0.1	3.1 ± 0.1
	% ^b^	0.7 ± 0.1	0.7 ± 0.1	0.6 ± 0.0	0.7 ± 0.1	0.7 ± 0.1	0.7 ± 0.1	0.7 ± 0.1	0.8 ± 0.1	0.8 ± 0.0 **	0.9 ± 0.1 **	0.6 ± 0.0	0.7 ± 0.0	0.6 ± 0.0	0. 7 ± 0.0
spleen	g ^a^	0.7 ± 0.1	1.1 ± 0.2	0.6 ± 0.1	1.1 ± 0.1	0.7 ± 0.1	1.2 ± 0.1	1.2 ± 0.0	1.2 ± 0.3	0.8 ± 0.1 **	1.3 ± 0.2	0.7 ± 0.1	1.2 ± 0.0	0.8 ± 0.1	1.3 ± 0.0
	% ^b^	0.2 ± 0.0	0.2 ± 0.0	0.2 ± 0.0	0.2 ± 0.0	0.2 ± 0.0	0.2 ± 0.0	0.3 ± 0.0	0.2 ± 0.0	0.3 ± 0.0 **	0.3 ± 0.1	0.2 ± 0.0	0.3 ± 0.0	0.3 ± 0.0	0.3 ± 0.0
thyroid	g ^a^	0.3 ± 0.1	0.4 ± 0.2	0.3 ± 0.1	0.3 ± 0.1	0.3 ± 0.1	0.4 ± 0.2	0.3 ± 0.0	0.4 ± 0.1	0.2 ± 0.1 *	0.4 ± 0.1	0.3 ± 0.1	0.3 ± 0.0	0.3 ± 0.0	0.3 ± 0.0
	% ^b^	0.1 ± 0.1	0.1 ± 0.0	0.1 ± 0.0	0.1 ± 0.0	0.1 ± 0.0	0.1 ± 0.0	0.1 ± 0.0	0.1 ± 0.0	0.1 ± 0.0	0.1 ± 0.0	0.1 ± 0.0	0.1 ± 0.0	0.1 ± 0.0	0.1 ± 0.0

* Represents a significant difference at the *p* < 0.05 level compared with the control group. ** Represents a significant difference at the *p* < 0.01 level compared with the control group. (n) Animal number (^a^) Weight (g); unit of absolute weight of organ. (^b^) Ratio (%); Relative ratio of organ weight to body weight.

**Table 5 ijerph-19-10413-t005:** Histopathological findings of SD rats orally administered AQ for 90 days.

Organ and Tissue Findings		End of Treatment Period (mg/kg)										End of Recovery Period (mg/kg)			
		0.00 mg/kg		1.36 mg/kg		5.44 mg/kg		21.76 mg/kg		174.08 mg/kg		0.0 mg/kg		174.08 mg/kg	
		♀	♂	♀	♂	♀	♂	♀	♂	♀	♂	♀	♂	♀	♂
Liver	hypertrophy	0/10	0/10	0/10	0/10	0/10	0/10	0/10	0/10	10/10 **	0/9	0/5	0/5	0/5	0/5
Kidney	hyaline droplets	0/10	0/10	0/10	6/10 *	7/10 **	8/10 **	10/10 **	9/10 **	9/10 **	7/9 **	0/5	0/5	2/5	0/5

* Represents a significant difference at the *p* < 0.05 level compared with the control group. ** Represents a significant difference at the *p* < 0.01 level compared with the control group. Each value represents the number of animals with each finding/the number of surviving animals of the same sex.

**Table 6 ijerph-19-10413-t006:** Graded results of eosinophilic glassy droplet scoring in the kidneys of rats at the end of the AQ trial.

Grading of Histomorphological Findings	Group									
	0.00 mg/kg		1.36 mg/kg		5.44 mg/kg		21.76 mg/kg		174.08 mg/kg	
	♀	♂	♀	♂	♀	♂	♀	♂	♀	♂
normal	10/10	10/10	10/10	4/10	3/10	2/10	0/10	1/10	1/10	2/9
minimal	0/10	0/10	0/10	6/10	0/10	0/10	0/10	0/10	0/10	0/9
mild	0/10	0/10	0/10	0/10	7/10 **	8/10 **	10/10 **	9/10 **	9/10 **	7/9 **
moderate	0/10	0/10	0/10	0/10	0/10	0/10	0/10	0/10	0/10	0/9
severe	0/10	0/10	0/10	0/10	0/10	0/10	0/10	0/10	0/10	0/9

** Represents a significant difference at the *p* < 0.01 level compared with the control group. Each value represents the number of animals with each finding/the number of surviving animals of the same sex.

**Table 7 ijerph-19-10413-t007:** Results of immunohistochemical grading of renal **α**-2u globulin in male rats at the end of the AQ trial.

Grading of Histomorphological Findings	Group				
	0.00 mg/kg	1.36 mg/kg	5.44 mg/kg	21.76 mg/kg	174.08 mg/kg
normal	3/10	1/10	1/10	2/10	3/9
minimal	3/10	6/10	7/10	2/10	6/9
mild	4/10	3/10	2/10	6/10	0/9
moderate	0/10	0/10	0/10	0/10	0/9
moderately severe	0/10	0/10	0/10	0/10	0/9

Each value represents the number of animals with each finding/the number of surviving animals of the same sex.

## Data Availability

Derived data supporting the findings of this study are available from the corresponding author Y.S. on request.

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
