# Peer review of "Acute and Subchronic Oral Toxicity of Anthraquinone in Sprague Dawley Rats"

_ijerph, 2022, doi:10.3390/ijerph191610413_

Round 1

Reviewer 1 Report

This manuscript entitled "Acute and subacute oral toxicity of anthraquinone in Sprague-Dawley Rats" by Qu Jingjing et al, mainly focusing on the subacute oral toxicity of AQ on SD-rat through oral route for 90 days. The results shown toxicity at certain doses (not in dose-dependent manner), with changes in body weight, and organ toxicity (based on biomarkers). The current manuscript is quite well written, however there are some info need to be clarify/improved, as follow:

1. Introduction: The rationale of test AQ for subacute toxicity is weak. The author describe the Chinese tea with AQ level exceeding 0.02mg/kg failed the export permit. However, what is the amount of AQ in the food / tea that human exposed daily? For instance, Chinese tea drinker take 200mL of tea daily and how much AQ will be found or cumulative in the body? At least make the rationale of the research stronger. In addition, in introduction, the acute oral toxicity mentioned is >5000mg/kg in rat. In that case, what is the rationale author test 2000mg/kg?

2. Line 78-81 should be mentioned in methodology, not introduction.

3. Line 105, please provide justification on 2 times gavage within 24 hours for acute toxicity test.

4. Line 118, author can please provide formulation on how to obtain the mentioned doses for subchronic toxicity part.

5. It is good if author can mention the outbreed SD-rat in section 2.2 (I hope the source of SD-rat is reliable as SD-rat is outbreed and used for toxicity study).

6. Figure 1 is unclear on labelling. 

7. Line 248, the mouse died after how many days of exposure? For subacute, the gavage is daily for 90 days, or 2 times in 24h for 90 days? The details are not mentioned.

8. Please explain the satellite control clearly. How the satellite control has lower body weight than the treatment group in male?

9. For table-2, it will be good if author can group the biomarkers according to organ.

10. For hormones in table-3, it is unclear why author only study thyroid, but not other reproductive hormones since author study sperm and vaginal toxicity as part of study.

11. What does the arrow in Figure 4 representing? Hyaline droplet? Please mention in Figure legend.

12. Please provide references for table 6 and table 7, for the grading. How the author define normal, minimal, mild, moderate, severe for grading? 

13. Discussion: The main issue of the manuscript is in discussion. There is no citation in discussion to compare with previous studies as reported in introduction. The authors report mainly on what they found, but did not discuss why certain toxicity happened in male but not female, vice-versa. This section need to improve.

14. Conclusion: Quite confusing on acute toxicity more than 2000 mg/kg, or 5000 mg/kg.

15. It is unclear why the age / body weight of the rats used in acute and chronic are different. 

16. For numerical value in all tables, kindly standardize to 1 or 2 decimal places.

17. For statistical analysis, the analysis between treatment and satellite groups was done, however, the number of rats used are different. Analysis shouldn't be performed in same way as other treatment group, or at least authors clarify more on the type of test (normality, non-parametric). 

Reviewer 2 Report

This is a straightforward manuscript where the authors exposed rats to Anthraquinone according to regulatory guidelines in order to determine its acute and chronic toxicity. While the manuscript is generally easy to read and understand, a few sentences in the discussion are deficient and should be rewritten. The justification for the dose selection provided on L.104-105 (LD50>2000mg/kg according to the manufacturer) appears a bit weak in light of the LD50 information provided in L. 42-48. Instead, the authors may want to mention OECD 423 Guidelines as the rational for dose selection. As for the acute toxicity of AQ which was compared to available information, chronic toxicity NOAEL should also be compared and discussed in light of previously published data. This manuscript should be suitable for publication once these issues and the comments bellow are addressed.

L. 35: Is “paper drying agent” correct? AQ is more commonly described as a pulping catalyst in paper production.

L.42-46: Please provide the species for different LD50s and proper references.

L. 67: Was the 31.3 mg/kg BW directly calculated from food consumption by the author of the cited study or estimated by you based on rat average food consumption?

L. 76-77: The sentence “we also added an acute toxicity test as the dose of AQ was different from the previously reported values” is not clear. Do you mean that the NOAEL value was different from the previously reported values?

L. 83-87: Please specify AQ batch number, if available. AQ is insoluble in water, so you may want to specify that insoluble AQ was resuspended in 1% carboxymethyl cellulose sodium. (While this appear to have been considered later at L. 144-128 you may still want to state this fact explicitly here).

L. 112: It may be preferable to add “examination” after  “gross anatomy”.

L. 176-181 and 289-291: Were urinary metabolites normalized to creatinine or specific gravity?

L. 218: I am assuming that Kruskall-Wallis was only used when endpoints did not meet homogeneity assumption, correct? Also repeated measure ANOVA should have been used to assess rat weight gains.

L. 226-240: Please specify if any significant differences were observed between the top, middle and bottom of the suspension.

L. 248: Was an autopsy performed on the dead rat to identify the cause of death (ex.: lung aspiration of the dosing material)?

L. 306: “symbolic” just seems wrong. You may want to use “significant” or meaningful”.

L 334-339: Please add an immunohistochemistry sub-section describing the method in the Material and methods section.

L. 370-371: This sentence is not exactly correct as the AQ dose administered was too low to determine a maximum tolerable dose. This needs to be better discussed.

L. 373: Please spell out “GHS”.

L 379-381: This sentence needs to be better written as it seems to claim that there is both difference and no difference in weight gains in the 174 mg/kg BW treatment group. Did you want to say that there were no difference in the satellite treatment groups? The next sentence (L. 381-384) also needs to be better explained.

L. 393: “at the end of the staining” seems very strange, did you mean “at the end of the dosing period”?

L. 406-407: Instead of “the effects of AQ on the liver and kidney of rats can be recovered” it may be preferable to say that “rats can recover from the effects of AQ on the liver and kidney”. The same comments apply to “the effect of AQ on the RET% in rats could be recovered” (L. 399-400). (There is something odd with the syntax of these sentences. Someone usually recover from something, and when something is recovered, is usually means that it is found or retrieved.)

L. 429: “were basically normalized” is just strange. You may want to mention that the pathological histological changes noted at the end of the AQ exposure period were not observable at the end of the recovery period.

L. 488: Please correct “Nomenclsture” and “Micestandard”.

Round 2

Reviewer 1 Report

The manuscript has been revised according to the comments earlier. It looks better now.

However, there are still some minor comments as follow:

1. The reply to the earlier comment no.3 (justification on 2 times gavage within 24h for acute toxicity test), the reply from authors sound acceptable. Kindly include the sentences (4 h interval each time) into the text under section 2.3.

2. For the earlier comment no.8, the reply from authors is not scientifically sound. If the weight of male rats in satellite control is lower, it should then normalized individual rat to before treatment, make y-axis percentage of body weight changes. Unlike female rats, the lower body weight in satellite group also can increase and comparable with other groups.

3. The authors changed the grouping to "organ" but it is biomarkers. The authors misunderstand my comment. If the grouping based on organ is difficult due to share biomarkers, then just remain. But changing the biomarker to organ is wrong.

4. I not understand why the authors deleted the sentences 108-110, which I think is highly important to put the animal ethical approval number/code from your institution.

I recommended for publication once the above comments are addressed. 
